# A Study on the Surface Quality of Selective Laser Melted Cylindrical- and Parallelepipedic-Shaped Inner Structure

**DOI:** 10.3390/ma16134649

**Published:** 2023-06-28

**Authors:** Yuyi Mao, Xinfeng Lv, Xiaodong Shen

**Affiliations:** 1College of Materials Science and Engineering, Nanjing Tech University, Nanjing 211816, China; lwtgzyyx2023@163.com; 2State Key Laboratory of Materials-Oriented Chemical Engineering, Nanjing Tech University, Nanjing 211816, China; 3Wuxi Institution of Inspection, Testing and Certification, Wuxi 214028, China

**Keywords:** selective laser melting (SLM), parallelepipedic-shaped inner structure, cylindrical-shaped inner structure, overhanging surface, improved model

## Abstract

A systematic study was conducted to investigate the distinct mechanisms involved in the formation of the inner surfaces of cylindrical and parallelepipedic-shaped structures. The surface roughness, flatness, and sinking distance were used as key indices to measure the quality of overhanging surfaces, while the surface flatness and roughness were used to evaluate the quality of the side and bottom surfaces of the inner hole. The inner surface morphology was observed using a scanning electron microscope and a white light interferometer. The test results show that the quality of the overhanging surface had a significant impact on the quality of the parallelepipedic-shaped inner hole. In contrast, the cylindrical-shaped inner hole had a shorter but more uniformly distributed overhanging surface, resulting in a different behavior of the overhanging and side surface quality. An improved model of the overhanging surface was established by combining all of the above results and comparing them with the traditional Euler Bernoulli beam model. The factors affecting the quality of the overhanging surface were analyzed, and measures for improving the quality of the overhanging surface during the SLM manufacturing process were proposed.

## 1. Introduction

With the increasing demand for energy consumption, the use of inner structure parts has become a focus of attention as it can not only reduce the energy output, but also enhance the properties of printed parts [1,2,3,4]. Compared to other methods such as material alternatives or machine redesign, inner structure parts are more intuitive and cost-effective. However, traditional manufacturing methods have limitations in machining inner holes and controlling the quality of the printed parts due to uncontrollable process parameters. In the past, many researchers had to compromise their design to meet manufacturing requirements instead of utilizing their best performance [5,6,7].

Fortunately, with the maturation of additive manufacturing technology in recent years, researchers have refocused on inner structure parts, as it has become possible to manufacture complex inner shapes using methods such as selective laser melting (SLM). SLM is a bottom-up process that uses a laser to melt metal powder according to the 3D model provided. The entire process is conducted under a protective gas to prevent metal oxidation [8,9,10,11,12,13,14,15].

As SLM differs significantly from traditional subtractive manufacturing methods, it is considered one of the most promising technologies for fabricating inner structure parts. Wang et al. investigated the impact of the laser power, scan speed, and inclination angle on the overhanging surface quality of printed parts [16]. Chen et al. attempted to improve the overhanging surface quality by adjusting the process parameters and found that decreasing the laser energy density could reduce the surface roughness while increasing the porosity [17]. Patterson et al. used a finite element analysis to simulate the temperature fields and distortion of the overhanging surface during the printing process [18]. Wu et al. discovered that lower inclination angles resulted in a decrease in the overall overhanging layer performance [19]. Finally, Duan et al. printed a cylindrical-shaped inner hole and found that laser power significantly impacted the overhanging surface quality, with increasing laser power leading to an increase in the roundness value, but a decrease in the dimension accuracy [20].

In contrast to the non-support method of printing inner holes, researchers have explored the use of support structures to improve the quality of inner holes. Kajima et al. found that adding support structures to fabricate overhanging surfaces significantly improved the fatigue strength compared to printing without support structures. This was due to the reduction in distortion and the increased cooling rate of the overhanging layer printed with support [21]. Zhang et al. incorporated cuboids into conventional block-type support structures and used the Taguchi method to optimize the structure, resulting in well-controlled distortion [14]. Leary et al. employed voxel-based cellular automata to generate support structures during printing, which enabled the direct fabrication of topology-optimized geometries [22]. Zhang et al. also developed branch-type support structures that improved cost savings and strength during printing [23].

Although the use of support structures can improve part performance, it has drawbacks such as time and material wastage, difficulty in removing support from small holes, and secondary damage during removal [24]. Therefore, this work aims to print the inner structure without any support structures. To further study the inner structure fabrication via SLM, this work compares the two most representative types of inner structures, cylindrical- and parallelepipedic-shaped holes, and provided a different forming mechanism to address the previous gap in understanding the formation of inner structures.

## 2. Materials, Instruments, and Experiment

### 2.1. Materials

The TC4 (Ti-6Al-4V) powder, produced using the plasma rotating electrode process method, was generously provided by Shenzhen Minatech Co., Ltd., Shenzhen, China. A scanning electron microscope (SEM) image of the TC4 powder can be seen in Figure 1, and further relevant information can be found in Table 1.

### 2.2. Instrument

The SLM equipment used in this experiment was provided by Nanjing University of Aeronautics and Astronautics. The model of this instrument was RAP-IV, which was made by their own lab. The primary process parameters are listed in Table 2. The design drawing of the formed part, featuring both the cylindrical and parallelepipedic inner characteristic structures, is depicted in Figure 2. It should be noted that three samples were printed with cylindrical and parallelepipedic inner structures to lower the impact of contingency in the experiments.

### 2.3. Forming Quality Test

Following printing, the samples were carefully cut from the center of the hole. To minimize any potential damage caused by cutting, a low-speed wire cutting machine (Bodor S3015, Jinan, China, with laser power 1200 W) was utilized to ensure precision. Trilinear coordinate measuring instrument was employed in this work to measure surface flatness. Triangulation technique using HP-L-20.8 Scanner Head was also used to ensure the overall precision of the measured results (Hewlett-Packard Development Company, Palo Alto, CA, USA). Samples were scanned to obtain point cloud data using ultrafine mode. Laser probe direction was changed during this process to increase the result precision. Noisy points were deleted, and the final results were compared with the design drawing. Specified surface flatness value was obtained accordingly. All calculations were finished using Polyworks, 2016 (InnovMetric Corporation, Québec, QC, Canada). The sinking distance of the overhanging surface was also calculated using this method. Additionally, the roughness was measured using Surface Roughometer, and Ra was employed in this work. The sample length was 2.5 mm, which is in accordance with JB/T 7051-2006 with 5 intervals. Each interval length was 1 µm. The length of the accelerating and decelerating parts was 1.25 mm. All the surfaces were measured in the direction along the inner structure. Three times measurement was employed in this work to lower the impact of error in the measurement process. Average value was calculated separately. To gain a better understanding of the phenomenon, SEM (Carl Zeiss, Sigma300, Oberkochen, Germany) was also used to obtain the surface morphology. A 3D surface profilometer (RTEC, MFD-F profilometer 2207, San Jose, CA, USA) was also used in this work to study the inner surface quality. The magnification was 20× and the numerical aperture was 0.4.

## 3. Results and Discussion

The maximum sinking distance of the cylindrical- and parallelepipedic-shaped inner structures on the overhanging surfaces was measured and recorded in this study. To minimize the impact of measurement errors, three different dimensions of inner holes were all measured on the three printed samples. The arithmetic mean value was also calculated, and the deviation was also presented on the surface roughness, sinking distance, flatness, and roundness values shown in the following tables. The results, presented in Table 3, clearly show that the sinking distance of the overhanging surface of the parallelepipedic-shaped inner structure was longer compared to that of the cylindrical-shaped inner structure. This can be attributed to the longer distance of the parallelepipedic-shaped overhanging surface. Interestingly, as the dimension of the parallelepipedic shape increased, the sinking distance of the overhanging surface showed a significant upward trend, confirming the previous explanation.

However, an opposite trend was observed for the sinking distance of the cylindrical-shaped inner structure, which did not show a significant increasing trend. In fact, it even decreased when the diameter of the cylindrical hole reached 3.0 mm. To investigate this phenomenon, SEM was used to observe the morphology of the overhanging surfaces of the 2.5 mm parallelepipedic-shaped and cylindrical-shaped inner structures, as shown in Figure 3.

From the SEM images in Figure 3, it can be observed that the sinking distance of the overhanging surface of the parallelepipedic-shaped inner hole was mainly caused by the lack of support on the bottom layer, which was loose powder instead of solidified metal. The large gap in the heat conductivity rate between the solidified metal and loose powder led to a longer solidification time for the molten pool on the overhanging surface, resulting in a longer sinking distance. The capillary force between the molten pool and the powder in the hole increased the weight of the molten pool, further aggravating the sinking distance. The sinking molten pool morphology and powder accumulation in the center provided evidence for this explanation.

As for the cylindrical-shaped inner structure, the overhanging surface sinking distance was caused by two reasons. One was the sinking molten pool, which was the same as that of the parallelepipedic-shaped overhanging surface. The other reason was the accumulation of powder under the overhanging surface due to the narrow space left in this position. As the diameter of the cylindrical-shaped inner hole increased, the distance of the overhanging layer had a limited increase due to the gradual shape change, while a wider space was provided at the same time. This explains why the sinking distance of the 3.0 mm cylindrical-shaped inner hole showed a decreasing trend in the measured data.

In this section, we present an analysis of the roughness, flatness, and roundness of the inner holes with parallelepipedic and circular shapes. The results are summarized in Table 4 and Figure 4. We find that the roughness of the bottom surface of the parallelepipedic inner hole (from 6.45 μm to 6.57 μm) is much lower than that of the circular inner hole (from 7.25 to 7.18 μm) due to the latter’s step effect. The side surface roughness of the circular inner hole (from 11.45 to 12.11 μm) is higher than that of the parallelepipedic inner hole (from 9.60 to 9.35 μm) due to the overhanging structures on the side surfaces, which cause heavier powder bonding which can be seen in Figure 5.

We also investigate the flatness and roundness of the two inner hole shapes. The flatness of the overhanging surface of the parallelepipedic inner hole (from 0.102 to 0.126) decreases as the length of the side increases (from 2.0 mm to 3.0 mm), but no such trend is observed for the side or bottom surfaces (from 0.085 to 0.080 and from 0.073 to 0.072, respectively). The roundness of the inner hole is not significantly affected by the diameter, but rather by the gradual change in the shape of the circular inner hole.

Based on these findings, we conclude that different optimization strategies should be employed for different inner hole shapes. For both circular and parallelepipedic inner holes, the length of the overhanging surface is the main factor affecting the surface quality. However, the traditional cantilever beam model used to simplify the machining of the overhanging surfaces assumes that the surface is horizontal, whereas our results show that the surface trends downward during selective laser melting (SLM) processing. Therefore, we propose a modified cantilever beam model based on the Euler–Bernoulli beam theory and the actual SLM process to improve the accuracy of the overhanging surface processing. Our results also support the hypothesis that powder sticking is the main cause of the side surface roughness, as confirmed by the side surface topography.

In Figure 6a, the model in the design process is depicted, revealing that the presence of an arc in the cylindrical structure causes some entities to exceed the arc in the layering process, resulting in a small overhanging surface. However, the overhanging feature appears larger in the schematic diagram. In reality, the thickness of each layer is only 10 to 200 μm, so the hanging distance is significantly less than shown in the diagram. Figure 6b shows an enlarged view of the n-th layer, which illustrates that the actual processing of the overhanging surface should exhibit a drooping shape with a certain radian, instead of being horizontal, as shown by the blue dotted line in the traditional model.

The surface is challenging to characterize mathematically; thus, it is simplified by connecting the front and end of the actual machining plane and defining the angle with the horizontal direction as *θ*, which is related to the properties of the molten pool. When the molten pool is too heavy and the sinking phenomenon is severe, *θ* increases, and the whole overhanging surface drops sharply. Here, *x* represents the total length of the n-1 layer, *b* represents the length of the n-th layer, and h(*x*,*θ*) represents the actual overhanging height. Solving for h(*x*,*θ*) reveals the displacement of the overhanging surface in the horizontal or vertical direction, which is closely related to the thickness of the material, the sinking angle of the overhanging surface, and the distance of the processed layer.
(1)$$h\left(x,\theta \right)=t-\left(x\cdot cos^{-1}\theta +t\cdot \sin\theta +b\cdot \cos\theta \right)\cdot \sin\theta +x\cdot \tan\theta \;=t\cdot cos^{2}\theta -b\cdot \sin\theta \cdot \cos\theta$$

The displacement in the *Y* direction caused by its drooping is as follows:(2)$$Y\left(x,\;\theta \right)=l\cdot sin\theta =\left(t\cdot sin\theta +b\cdot cos\theta +x\cdot \cos^{-1}\theta \right)sin\theta$$

Similarly, such drooping also leads to a certain position offset in the *X* direction, and the displacement can be expressed as follows:(3)$$Y\left(x,\;\theta \right)=l\cdot sin\theta =\left(t\cdot sin\theta +b\cdot cos\theta +x\cdot \cos^{-1}\theta \right)sin\theta$$

To resist the tendency of expansion and contraction of the TC4 powder during processing and cooling, a certain force is required by the formed part to resist residual stress, as shown in Figure 7.

*F*_1_ represents the tensile stress required by the structure of the overhanging surface to resist residual stress during the actual processing and cooling of the titanium alloy, which can be expressed as follows:(4)$$F_{1}=\sigma_{m}mtb$$
where $\sigma_{m}$—yield stress of the TC4 powder (N), m—the ratio between the depth of powder in the molten pool and the thickness of the layer in laser processing, and *t*—layer thickness (mm).

The supporting force *F*_2_ provided by the supporting structure is as follows:(5)$$F_{2}=F_{1}\cdot \cos\theta =\sigma_{m}mtb\cdot \cos\theta$$

The momentum *M*_2_ provided by the supporting structure is as follows:(6)$$M_{2}=\sigma_{m}mtb\cdot \cos\theta \cdot \frac{h\left(x,\theta \right)+t\cdot \cos\theta }{2}$$

Additionally, the deformation caused by the axial compression of the support structure δ(x) can be used for calculation as follows: (7)$$\delta \left(x\right)=-\frac{F_{2}}{E}\int_{0}^{x}\frac{dx}{A\left(x\right)}$$
where *A*(*x*)—variable cross-sectional area (mm^−2^), and *E*—Young’s modulus of materials (Pa).

*A*(*x*) can be calculated using following formula: (8)A(x)=b·(h(x, θ)+t·cosθ)

Equation (9) can be obtained by taking it into the above formula as follows: (9)$$\delta \left(x\right)=-\frac{\sigma_{m}mt\cdot \cos\theta }{E}\int_{0}^{x}\frac{dx}{h\left(x,\theta \right)+t\cdot \cos\theta }=-\frac{\sigma_{m}mt\cdot \cos\theta }{E}\int_{0}^{x}\frac{dx}{t\cdot cos^{2}\theta -b\cdot \sin\theta \cos\theta +t\cdot \cos\theta }$$

The displacement of the *m*-th layer caused by the displacement of the *n*-th layer is as follows:(10)$$D_{mn}=\delta \left(x_{i}\right)=-\frac{\sigma_{m}mt\cdot \cos\theta }{E}\int_{0}^{x}\frac{dx}{t\cdot cos^{2}\theta -b\cdot \sin\theta \cos\theta +t\cdot \cos\theta }=\frac{\sigma_{m}mt}{E\left(t\cdot cos\theta -b\cdot sin\theta +t\right)}$$

Since the displacement can be accumulated, the x-direction displacement caused by the residual stress in the *m*-th layer can be obtained, which can be expressed as follows:(11)$$D_{m}=\sum_{n=m}^{T}D_{mn}$$
where *T*—total number of layers of the processed sample.
(12)$$W\left(m,\theta \right)=\alpha D_{m}+\beta X\left(x,\theta \right)$$
where *α*, *β* are coefficients and 0 < *α*, *β* < 1.

In addition to the horizontal displacement, there is also vertical displacement of the suspension surface. The Euler–Bernoulli beam theory is used to calculate it as follows:(13)$$\frac{M_{2}}{\mathrm{EI}}=\frac{d^{2}\Delta }{dx^{2}}$$

In this formula, the suspension section can be simplified as a rectangular plane and can be obtained as follows: (14)$$I\left(x,\theta \right)=\frac{1}{12}b\cdot cos\theta \cdot {\left(h\left(x,\theta \right)+t\cdot \cos\theta \right)}^{3}$$

Then, the above formula can be reduced to the following:(15)$$\frac{d^{2}\Delta }{dy^{2}}=\frac{\sigma_{m}mt}{E\cdot {\left(h\left(x,\theta \right)+t\cdot \cos\theta \right)}^{2}}$$
(16)$$\Delta \left(y\right)=B_{mn}$$
(17)$$B_{m}=\sum_{n=m}^{T}B_{mn}$$

Combined with the previous deformation caused by the molten pool sinking, it can be found that the displacement of the final overhanging surface in the vertical direction *Q* (*m*, *θ*) is as follows: (18)$$Q\left(m,\theta \right)=\gamma B_{m}+\varepsilon Y\left(x,\theta \right)$$

Similarly, here, *α*, *β* are coefficients, and 0 < *α*, *β* < 1.

The modified overhanging surface processing model takes into account the weight of the molten pool, which affects the initial position of the cantilever beam, and subsequently affects the overall displacement distance of the overhanging surface, thereby influencing the processing quality of the overhanging surface. Additionally, the original beam shape changes due to the sinking of the molten pool under the action of gravity, affecting the displacement in the *X* and *Y* directions. The horizontal beam in the traditional model ignores the displacement caused by the subsidence of the suspended surface, which affects the accuracy of the subsequent model description. The observed morphology of the overhanging surface verifies that the modified overhanging surface processing model is more consistent with the actual situation of the overhanging surface manufactured via SLM.

To confirm the theory given above, an experiment was conducted with a lower layer thickness value of 0.08 mm, and with the other parameters all kept the same during the process. The parallelepipedic-shaped inner structure was studied in this section due to the fact that powder accumulation was more likely to affect the overhanging surface performance, which was mentioned above. The testing results can be found in Figure 8.

From the testing results, it can be found that with a lower layer thickness, the overall performance of the overhanging surface had an upward trend. With the increase in the inner structure dimension, the overhanging surface quality showed a more significant improvement with a lower thickness value, which further confirmed the explanation given above.

## 4. Conclusions

In this study, we compared the surface quality of inner structures with parallelepipedic and cylindrical shapes, particularly focusing on overhanging surfaces. Our findings reveal that the overhanging surface quality of the parallelepipedic-shaped inner structure (roughness value ranged from 16.73 to 19.95 μm; flatness value ranged from 0.102 to 0.126) was inferior to that of the cylindrical-shaped inner structure (roughness value ranged from 13.63 to 13.05 μm; roundness value ranged from 0.102 to 0.110) due to two main reasons. Firstly, the length of the overhanging layer was longer in the parallelepipedic shape, resulting in a sharp change in the shape compared to the smoother change in the cylindrical shape. Consequently, the layer under the overhanging surface of the cylindrical shape provided a higher support force during the printing process compared to the parallelepipedic shape. Secondly, the overhanging layer space of the cylindrical shape was smaller than that of the parallelepipedic shape, leading to the accumulation of extra powder and an increased sinking distance. Additionally, we found that the quality of the side inner surface of the cylindrical shape was lower than that of the parallelepipedic shape, primarily due to the extra powder bonding on the steps of the side surface. Based on our results, we modified the traditional beam model and compared it with the optimized model, using the obtained data and surface topography. Our results demonstrate that the optimized model is more reasonable and can produce superior quality overhanging surfaces in selective laser melting.

## Figures and Tables

**Figure 1 materials-16-04649-f001:**
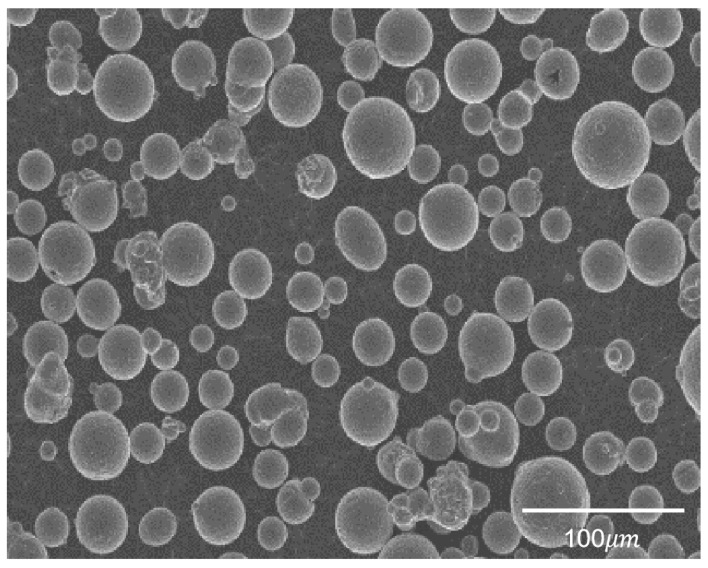
SEM image of TC4 powder.

**Figure 2 materials-16-04649-f002:**
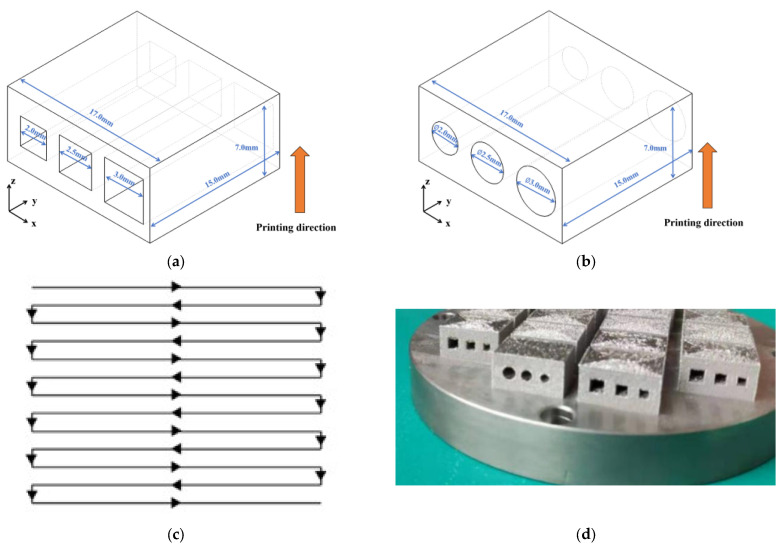
The design drawing of inner structure; (**a**) parallelepipedic-shaped inner structure, (**b**) cylindrical-shaped inner structure, (**c**) scan strategy, and (**d**) printed samples in this work.

**Figure 3 materials-16-04649-f003:**
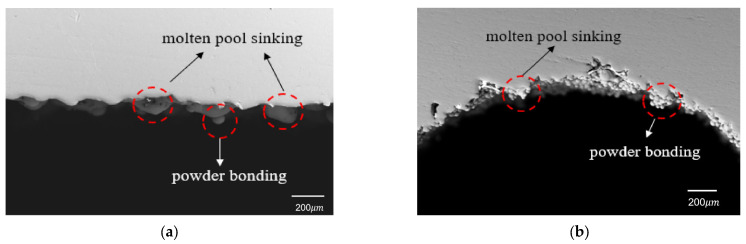
Overhanging surface morphology of 2.5 mm inner hole: (**a**) parallelepipedic shape and (**b**) cylindrical shape.

**Figure 4 materials-16-04649-f004:**
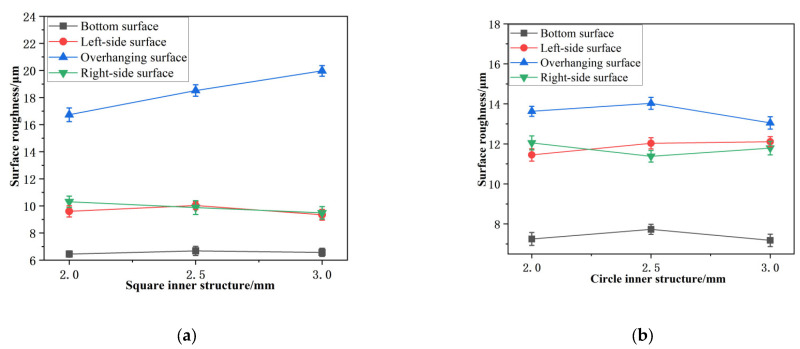
Variation of surface roughness of inner structure: (**a**) parallelepipedic-shaped structure and (**b**) cylindrical-shaped structure.

**Figure 5 materials-16-04649-f005:**
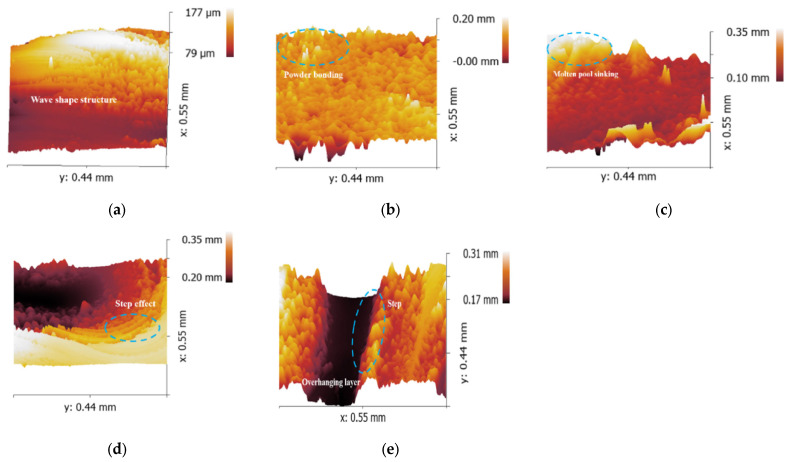
Inner structure morphologies: (**a**) bottom surface, (**b**) side surface, (**c**) overhanging surface of parallelepipedic-shaped structure and (**d**) side surface, (**e**) overhanging surface of cylindrical-shaped structure.

**Figure 6 materials-16-04649-f006:**
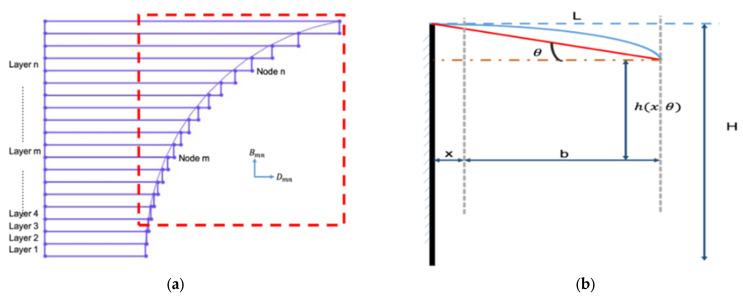
(**a**) Amplified cylindrical-shaped inner structure and (**b**) schematic diagram of the Euler–Bernoulli beam theory.

**Figure 7 materials-16-04649-f007:**
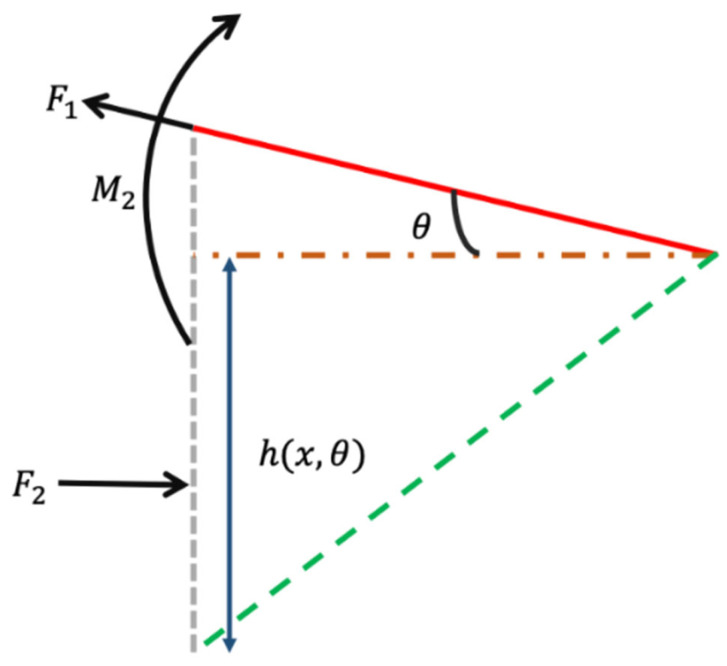
Amplified image on the beam position.

**Figure 8 materials-16-04649-f008:**
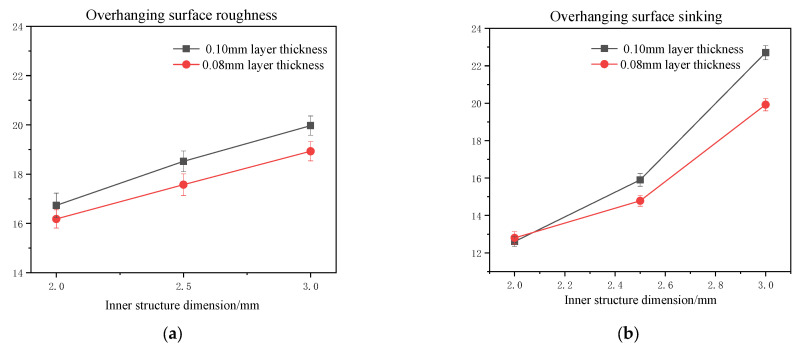
Comparison of overhanging surface roughness (**a**) and sinking distance (**b**) of parallelepipedic-shaped inner structure printed under 0.1 mm and 0.08 mm layer thicknesses, respectively.

**Table 1 materials-16-04649-t001:** Relative information of TC4 powder.

TC4	Sphericity	Flowability/s	Particle Size/μm			Element Content/%						
			D_10_	D_50_	D_90_	O	N	H	C	Al	V	Ti
Value	0.983	12.6	16.9	19.2	22.5	0.176	0.024	0.0064	0.010	5.74	3.83	Bal

**Table 2 materials-16-04649-t002:** Process parameters used in this work.

Laser Power	Scan Speed	Layer Thickness	Hatch Spacing	Spot Diameter	Scanning Strategy	Protective Gas
180 W	1000 mm/s	0.12 mm	0.10 mm	80 μm	Single	Argon

**Table 3 materials-16-04649-t003:** Sinking distance of parallelepipedic-shaped and cylindrical-shaped inner structure.

Inner Structure/mm	Parallelepipedic-Shaped/μm	Deviation/μm	Cylindrical-Shaped/μm	Deviation/μm
2.0	12.6	1.3	9.3	0.8
2.5	15.9	1.8	10.2	1.1
3.0	22.7	2.5	8.9	1.1

**Table 4 materials-16-04649-t004:** Flatness and roundness of parallelepipedic-shaped and cylindrical-shaped inner surface.

Inner Hole/mm	Flatness of Parallelepipedic-Shaped Inner Hole/mm								Cylindrical-Shaped Inner Hole/mm	
	Bottom	Deviation	Left Side	Deviation	Overhanging	Deviation	Right Side	Deviation	Roundness	Deviation
2.0	0.073	0.01	0.085	0.02	0.102	0.02	0.083	0.01	0.102	0.03
2.5	0.070	0.02	0.083	0.02	0.115	0.02	0.082	0.02	0.093	0.02
3.0	0.072	0.02	0.080	0.02	0.126	0.02	0.082	0.03	0.110	0.03

## Data Availability

Not applicable.

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
