# Peer review of "A Study on the Surface Quality of Selective Laser Melted Cylindrical- and Parallelepipedic-Shaped Inner Structure"

_materials, 2023, doi:10.3390/ma16134649_

Round 1

Reviewer 1 Report

The authors investigated the surface quality of square or circle shape overhanging inner structure of TC4 fabricated by SLM. The topic is suitable for this journal and the findings of this study may be interested in the readers of this journal. My major concern is the sample size is unclear. If the results are mean value SD should be written. Why didn't do statistical analysis? In addition, Table 4 and Figure5 are of the same result so one of them should be removed.

There are other minor concerns that should be considered for revision.

Minor concern

Materials

1.The TC4 should be described in more detail, like TC4 (T-6Al-4V).

2. Element content in Table 1 should include Ti as balance.

3. The method how to obtain the sinking distance is difficult to understand. The illustration or obtained 3D data should be added.

4. Authors must describe what kind of surface roughness was measured. Is it Ra? How many samples were used? How many times were the samples measured?

5. The methods how to measure the flatness and roundness was not described. They should be written along with their definitions.

Results

5. Authors must describe what kind of surface roughness was calculated. Table 4 should be revised too.

6. The results of statistical analysis should be described.

Conclusion

7. The conclusion should be revised based on the results of statistical analysis.

Reviewer 2 Report

JOURNAL: materials

Manuscript number: 2390229

Title: A study on the surface quality of SLMed square/circle shape inner structure

The title clearly describes the article and the abstract reflects the content of the article.

However, this study is concerned with internal volume surfaces. The terms "cylindrical and parallelepipedic shape" seem more appropriate.

This paper is concerned with the problem of shape deviations and significant roughness when making cavities in parts obtained by the LPBF process. The authors should justify the choice of shapes (and their size). These defects are very well described in the literature. They depend in particular on the lasering conditions and the lasering strategies. This last point should be clarified. The authors should indicate more precisely the objective of this paper.

1. Introduction

The introduction clearly states the problem being investigated and the objectives of this research.

2. experimental details

2.1 materials

The authors should specify the type of scanning electron microscope used and the observation parameters.

The legend to Figure 2 is difficult to read (measurement scale).

The authors should give details of the techniques used to obtain the results presented in Table 1: What type of equipment, what analysis parameters?

2.2 instrument

The authors should give the type of L-PBF machine used and the distribution of the specimens on the tray as the loading of the trays may affect the results.

The units in Figure 3 must be specified.

The orientation of the specimens concerning the lasering and construction axes should be specified. The lasering strategy is also important (layer thickness, hatch distance, unidirectional/bi-directional, rotation between layers, checkerboard strategy, contouring ...). It should be specified.

2.3 Forming quality test

The type of equipment (name of equipment, brand) for cutting and analysing the samples should be given.

What does "in outer view" mean in the sentence “a scanning electron microscope provided by Carl Zeiss, German was used to observe the morphology of the inner surfaces in outer view"?

3. Results

The overall shape of the holes made in cylindrical or parallelepiped shapes should be presented in a figure.

Figure 4: specify the size of the observed shape (2, 2.5 or 3mm?). Observations of the surface of the holes (in cross-section) would have made it easier to show surface defects. Perhaps it would have been possible to check the influence of the lasering strategy.

Figure 5: Specify the standard deviations on each point of the graph.

Table 5: the authors should specify the parameters chosen to describe the flatness and circularity of the inner square and circular surface.

Figure 6: This figure is difficult to read as such. The units of the X and Y axes need to be revised. The legend (colour) in the Z axis should be given.

In the last part, the authors propose a modified model to explain the shape deviations of hollow shapes in LPBF parts. The results obtained with this model should be compared with the experimental results.

This paper can be published after major corrections.

Round 2

Reviewer 1 Report

I have checked the revised version. I’m afraid to say that my major concerns have not been revised correctly. Figure 3 is not enough to show the exact sample number. Please describe the sample number in the manuscript, like n=X/group. In addition, I couldn’t find any mean value as well as SD. Please describe the mean value in the sentence clearly as well as in the table. What kind of statistical analysis was done? Please describe the method in the materials and methods section and the existence of significant difference in section. I cannot any results of statistical analysis, p-value.

Author Response

Dear Reviewer, 

Thanks for your time and consideration. Firstly, we must apologize for our misunderstanding on your suggestion. Indeed, we use arithmetic mean value in this work to lower the impact caused by the contingency happened in this work. As the powder bonding had a great impact on the surface quality, we could not have a further statistical analysis in this work due to powder random bonding during the experiment. However, as the whole variation trend was clear enough to make sense, we thought this work still had a great significance to improve the surface quality on different inner structures. Moreover, according to your suggestion,  standard deviation was given in the tables and the number of the samples was also given in the revised version

Thanks again for your suggestion and consideration.

Yuyi

Reviewer 2 Report

The paper deals with a well-known and explained problem in additive manufacturing: the realisation of hollow volumes inside the parts in additive manufacturing. They always present shape defects, and their surface is rough.

The authors have improved the quality of the paper, especially the description of the experimental part.

There are still corrections (e.g. figure 6; figure legend: colour level versus height) to improve the paper.

Author Response

Dear Reviewer,

Thanks for your time and consideration. According to your suggestion, we had a further discussion on Figure 6. The images captured by WLI was mainly used to show the surface quality shown before. We thought it made sense by given the deviation height in Y-axis. As for the color shown in this figure, it was calculated by the software which was hard to be given in the software.

Thanks again for your time and consideration. Hope to hear you again.

Yuyi